# Metal-organic Framework ZIF-67 Functionalized MXene for Enhancing the Fire Safety of Thermoplastic Polyurethanes

**DOI:** 10.3390/nano12071142

**Published:** 2022-03-29

**Authors:** Mei Wan, Congling Shi, Xiaodong Qian, Yueping Qin, Jingyun Jing, Honglei Che

**Affiliations:** 1School of Emergency Management and Safety Engineering, China University of Mining and Technology, Beijing 100083, China; wan1112mei@163.com (M.W.); qyp@cumtb.edu.cn (Y.Q.); 2Beijing Key Laboratory of Metro Fire and Passenger Transportation Safety, China Academy of Safety Science and Technology, Beijing 100012, China; wjxyqxd@hotmail.com (X.Q.); bdqyjjy@163.com (J.J.); sychehonglei@163.com (H.C.)

**Keywords:** metal-organic framework ZIF-67, MXene, fire safety, thermoplastic polyurethanes

## Abstract

In this work, a novel functionalization strategy for ZIF-67-modified layered MXene was proposed, aiming at improving the fire safety of thermoplastic polyurethanes (TPU). The ZIF-67@MXene was verified by microscopic morphology, elemental composition, functional group species and crystal structure, and then the successfully prepared ZIF-67@MXene was introduced into the TPU material. When ZIF-67@MXene content was only 0.5 wt%, the peak heat release rate, total heat release rate, peak smoke release rate, total smoke release rate, and CO yield of the TPU/ZIF-67@MXene composites were reduced by 26%, 9%, 50%, and 22%, respectively, compared with the pure TPU. The thermogravimetric tests showed that the residual char of TPU/ZIF-67@MXene composites was the most in all samples. In short, the high-quality carbon layer of TPU/ZIF-67@MXene composites acts as a physical barrier to the transfer of heat and toxic gases, greatly improving the flame retardant properties of the TPU polymer.

## 1. Introduction

Thermoplastic polyurethanes (TPU) are capable of adding great economic and social value to the production industry with excellent properties such as high strength, good toughness, cold resistance, aging resistance, and so on [1,2]. The TPU composites are usually applied in many fields, such as aerospace, automobile manufacturing, building, electronics and electrical appliances, etc., [3,4]. Nevertheless, TPU, like most polymers, is flammable in the presence of oxygen or air, and the combustion process is accompanied by large amounts of heat, toxic gases, and smoke [5,6,7]. Therefore, it is very important to improve the fire safety performance of TPU. With people’s increasing environmental awareness, halogen-free flame retardants are drawing more and more attention and are being developed rapidly.

At present, related studies have shown that the incorporation of nano-fillers is an effective means of improving the flame retardancy of TPUs [8,9,10]. Quan et al. introduced a method for preparing graphite nanosheet (GNP)-filled TPU nanocomposites. The thermal stability of TPU/GNP composites was significantly enhanced and the heat release rate (HRR) was significantly reduced [11]. Yu et al. proposed cetyltrimethylammonium bromide (CTAB) and tetrabutylphosphonium chloride (TBPC) modified Ti_3_C_2_ (MXene) ultra-thin nanosheets and the modified nanosheets can significantly reduce fire, smoke, and toxicity hazards of TPU polymers [12]. Since its discovery in 2011, MXene has been used in a wide range of applications in energy storage, communications, environment, catalysis, and so on [13]. MXene is a unique two-dimensional material with the formula M_n+1_X_n_T_x_, where M represents the transition metal (Ti, Zr, V, or Mo), X is carbon or nitrogen, and T represents the termination group (–O, –OH, and/or –F) [14,15]. Currently, more than half of the MXene studies have focused on Ti_3_C_2_Tx [16]. In general, MXene is prepared by selectively etching the A atomic layer of the MAX precursor using hydrofluoric acid to form graphene-like nanosheet structures [17,18,19]. However, fluoride salts and milder acid etching MAX phases are a better choice due to the dangers of hydrofluoric acid [20]. The reaction formulas are [21,22]:(1)$${\mathrm{Ti}}_{3}{\mathrm{AlC}}_{2}+3\mathrm{HF}={\mathrm{AlF}}_{3}+{\mathrm{Ti}}_{3}C_{2}+1.5H_{2}$$
(2)$${\mathrm{Ti}}_{3}C_{2}+2H_{2}O={\mathrm{Ti}}_{3}C_{2}{\left(\mathrm{OH}\right)}_{2}+{\;H}_{2}$$
(3)$${\mathrm{Ti}}_{3}C_{2}+2\mathrm{HF}={\mathrm{Ti}}_{3}C_{2}F_{2}+{\;H}_{2}$$

Due to the termination groups (–O, –OH, and/or –F), MXene nanoparticles are easily surface modified and grafted with chemically active groups, which can reduce the heavy accumulation of MXene flakes, increase the effective catalytic area, and improve their catalytic activity and stability [23,24,25]. In addition, MXene can play an excellent role in flame retardation with its layered barrier and metal ions to retard the combustion processes of polymers [26]. 

In recent years, a great deal of research has been devoted to developing superior properties of metal-organic frames (MOFs) in a variety of applications with gas storage, catalysis, and drug retardation [27,28,29]. MOFs, as an organic-inorganic hybrid material with two components acting as flame retardants, show significant advantages in terms of structure and composition [30,31,32]. Without the limitation of poor thermal stability, MOF is widely used in flame retardant areas. Fortunately, extensive studies have confirmed that the well-stabilized ZIF family provides a strong impetus for the flame retardant applications of MOFs [33,34]. Hou et al. designed and prepared nine pristine MOFs and 18 MOFs/polymer composites to investigate the effects of metal components and organic ligands on the flame retardant properties of the polymeric materials. Analysis of thermogravimetric (TG) and cone calorimetric test (CCT) results indicate that the CO-MOFs were superior to other MOFs with the same organic ligands and the ZIF series were better than the BDC series [35]. Xu et al. assembled water-soluble phenolic resin and ZIF-67 by electrostatic action on the surface of flexible polyurethane foam (FPUF), and the coating can effectively retard the release of gaseous toxic products during pyrolysis, providing a new solution to the problem of toxic fumes from FPUF [36]. Xu et al. prepared functionalized reduced graphene oxide (RGO) by adsorbing borate (ZIF-67/RGO-B) with Co-ZIF, indicating that the heat production and smoke production rate of the EP composites were significantly reduced [34]. Generally, there is no relevant report on the novel flame retardants combining the advantages of MXene and MOFs at present.

It is well-known that two-dimensional MXene nanosheets have a negative charge on the surface, while MOF has amounts of unsaturated sites. In this work, MOF-modified MXene was prepared and the flame retardants were introduced into TPU composites to improve the compatibility of MXene with the polymer and enhance the thermal stability and flame retardant properties of the composites. This work aims at providing a feasible and effective strategy for fabricating thermally stable, high flame retardant efficiency composites.

## 2. Experimental Section

### 2.1. Material

Titanium aluminum carbide (Ti_3_AlC_2_) was bought from Laizhou Kai Kai Ceramic Materials Co., Ltd. (Yantai, China). Lithium fluoride was provided by Shanghai Aladdin Bio-Chem Technology Co., Ltd. (Shanghai, China). Cobalt (Ⅱ) nitrate hexahydrate (Co(NO_3_)_2_·6H_2_O), methanol, and hydrochloric aids were purchased from Sinopharm Chemical Reagent Co., Ltd. (Shanghai, China). 2-Methylimidazole was supplied by Shanghai Macklin Biochemical Co., Ltd. (Shanghai, China).

### 2.2. Preparation of ZIF

2-Methylimidazole (3.2 g, 0.04 mol) was added to 250 mL of methanol and sonicated for 1 h (solution A). Co(NO_3_)_2_·6H_2_O (2.9 g, 0.01 mol) was dissolved in 250 mL of methanol and stirred for 10 min to fully dissolve (solution B). Next, solution A was poured into solution B. Then, the mixtures were stirred magnetically for 10 min and left to stand for 24 h. Finally, the precipitate was washed, filtered, dried at 80 °C for 24 h and then Co-ZIF was collected.

### 2.3. Preparation of MXene

LiF (2 g) was dissolved together in 40 mL of HCl and water solvent (V_HCl_: V_H__2O_ = 3:1). Ti_3_AlC_2_ powder (2 g) was slowly added to the LiF/HCl solution, and then kept at 40 °C with magnetic stirring for 24 h. Subsequently, the reaction solution was centrifuged to collect the precipitate (3500 rpm, 10 min), washed several times with deionized water, and centrifuged (3500 rpm, 5 min) until the mixture was pH neutral. Next, the resulting precipitate was sonicated with ethanol for 1 h, and then centrifuged (5000 rpm, 20 min) to obtain the precipitate. After that, deionized water was added to the precipitate and sonicated for 2 h to promote the delamination of MXene. Finally, the supernatant obtained by centrifugation (3500 rpm, 20 min) was used as a suspension of MXene nanosheets, which were processed in a freeze dryer for 48 h to obtain the MXene nanosheets.

### 2.4. Preparation of ZIF@MXene Hybrids

MXene/Co-ZIF materials were prepared by direct precipitation, as shown in Figure 1. In brief, 2-methylimidazole (3.2 g, 0.04 mol) and MXene (1 g) were added into 250 mL of methanol and sonicated for 1 h (solution A). Co(NO_3_)_2_·6H_2_O (2.9 g, 0.01 mol) was dissolved in 250 mL of methanol (solution B). Next, solution A was poured into solution B. Then, the mixture was stirred magnetically for 10 min and left to stand for 24 h. Finally, the precipitate was washed, filtered, dried at 80 °C for 24 h and then MXene/Co-ZIF hybrids was collected. The synthesis scheme is illustrated in Figure 1a.

### 2.5. Preparation of TPU Composite

TPU particles were co-blended with ZIF-67@MXene hybrids (0.5 wt%) or ZIF-67 (0.5 wt%) at 180 °C in a mixer machine. The homogeneously mixed TPU samples were then hot pressed at 180 °C for 10 min on a flat vulcanizing press. In addition, the pure TPU materials without flame retardants were prepared in the same way. The samples were named as TPU, ZIF-67/TPU, and ZIF-67@MXene/TPU.

### 2.6. Characterization and Analysis

#### 2.6.1. Energy Dispersive Spectroscopy (EDS)

The elemental composition and content of MXene, ZIF-67, and ZIF-67@MXene were analyzed by a Phenom XL G2 elemental microanalyzer (Eindhoven, The Netherlands).

#### 2.6.2. Scanning Electron Microscope (SEM)

The morphology of residual char after CCT was investigated by scanning electron microscope (Phenom XL G2, Eindhoven, The Netherlands). 

#### 2.6.3. X-ray Photoelectron Spectroscopy (XPS)

Elemental analysis was tested by the X-ray photoelectron spectroscopy (Thermo ESCALAB 250 Xi, ThermoFisher Scientific Inc., New York, America) with Al Kα radiation.

#### 2.6.4. Fourier Transform Infrared Spectroscopy (FTIR) 

The chemical structure was measured on the FTIR spectrum (NICOLET IS10, Thermo Nicolet Corporation, Madison, Wisconsin, America). The range of the infrared spectrum was 400–4000 cm^−1^, and the number of scans was 64 times.

#### 2.6.5. X-ray Diffraction (XRD)

The structural form of MXene, ZIF-67 and ZIF-67@MXene were conducted using a D/Max-3c X-ray powder diffractometer (D8-ADVANCE, Bruker Inc., Karlsruhe, Germany) for Cu Kα radiation. The diffraction angle was collected in the range of 10–90° at a speed of 4° min^−1^.

#### 2.6.6. Thermogravimetric Analysis (TGA)

The thermal degradation behavior of TPU composites was studied by using the NETZSCH synchronous thermal analyzers (STA 449 F5, Germany natch instrument manufacturing Inc., Selb, Germany) at a heating rate of 15 °C min^−1^ under air atmosphere or nitrogen atmosphere. The range of temperature was from 50 to 900 °C.

#### 2.6.7. Cone Calorimeter Test (CCT)

The CCT was performed using an FTT Cone calorimeter (Fire Testing Technology (FTT) Inc., East Grested, Britain) under an external heat flux of 35 kW m^−2^. The specimen size was 100 mm × 100 mm × 3 mm. 

#### 2.6.8. Raman Spectrometer

The chemical structure of residual char after CCT was observed by Raman Spectrometer (Renishaw inVia, Renishaw Inc., London, Britain). The wave numbers tested ranged from 500 cm^−1^ to 2000 cm^−1^ with incident wavelength of 532 nm, which was tested twice for each sample.

## 3. Results and Discussion

### 3.1. Characterization of ZIF-67 and ZIF-67@MXene

In Figure 1a, Ti_3_AlC_2_ is a large stacked layered structure. After etching and delamination of the bulk Ti_3_AlC_2_ with LiF and HCl, the elemental Al layer was successfully removed and Ti_3_C_2_T_x_ was a weakly stacked structure, which was dispersed into a monolithic layer structure after sonication. The terminal groups of MXene nanosheets (–O, –OH and/or –F) are easily surface modified and grafted with chemically active groups, and the Co metal ions of ZIF-67 nanoparticles have a large number of unsaturated sites, which interact with each other to form Co@MXene hybrids. The scanning electron microscopy (SEM) was first tested to examine the morphology and microstructure of the as-prepared ZIF-67 and ZIF-67@MXene hybrids. As shown in Figure 1b,c, there was a flat and smooth surface with well-defined edge angles for ZIF-67, which was composed of regular micron-sized particles of a dodecahedral rhombic morphology. The structure of ZIF-67@MXene hybrids was exhibited in Figure 1d,e, where the dodecahedra were dispersed and attached to the MXene lamellae, indicating that the modification was successful. By the way, the edges of ZIF-67 were blurred due to the interaction between the ZIF-67 and MXene, while the presence of the MXene favorably contributed to the smaller size of the ZIF-67.

Furthermore, 60 grains were randomly selected on the SEM graph of ZIF-67 and ZIF-67@MXene to measure the particle size distribution, as shown in Figure 2. The sizes of ZIF-67 particles range from 0 to 2.4 μm and are mainly concentrated in 0.9–1.2 μm, indicating that the self-made ZIF-67 particles are large. In addition, the ZIF-67 particle sizes in ZIF-67@MXene range from 0 to 0.7 μm, and the percentage of 0.2–0.3 μm sizes was 43.3%, indicating that MXene contributed to the formation of smaller nanoparticles of ZIF.

The elemental composition and content of as-synthesized ZIF-67 and ZIF-67@MXene were characterized by energy dispersive spectroscopy (EDS), as shown in Figure 3a. The peak of titanium and fluorine was observed, which confirmed the existence of MXene in the ZIF-67@MXene hybrids. The atomic percent values of C, N, O, F, Co, and Ti were 17.1, 18.2, 17.6, 6.3, 33.5, and 7.3, respectively. The atomic percent value of C, N, and Co was lower than the corresponding value of ZIF-67 due to the existence of MXene. Due to the use of fluoride-based etchants, the exposed M surface of the MXene layer was always capped with –F, –O, and –OH group. Thus, the elemental content of O was increased in ZIF-67@MXene hybrids. 

The synthesis of ZIF-67@MXene hybrids was demonstrated by the elemental crystal structure analysis by X-ray photoelectron spectroscopy (XPS). The C 1s, N 1s, Ti 2p, O 1s, F 1s, and Co 2p peaks of ZIF-67@MXene appeared at 285, 399, 458, 531, 684, and 782 eV, respectively. The XPS bands of Co 2p, C 1s, and N 1s of ZIF-67 are shown in Figure 2a,b The two characteristic peaks of ZIF-67 were Co 2p2/3 and Co 2p1/2 with binding energies of 781.1 eV and 796.68 eV, respectively, indicating that Co 2p spectra had Co^3+^ and Co^2+^ ions, and there were two satellite peaks nearby [37]. In addition, the high-resolution spectrum of N 1s was fitted with three peaks at 398.74, 402.12, and 406.10 eV, corresponding to the pyridine nitrogen and, graphitic nitrogen and oxidation nitrogen, respectively [38]. Moreover, the modification of MXene by ZIF-67 resulted in the appearance of the C–Ti peak at 281.24 eV in the C1s band [39]. Furthermore, in the high-resolution spectrum of Ti 2p, the four main peaks were identified at 455.14, 457.90, 460.80, and 463.55 eV, corresponding to Ti–C, Ti–O/F, Ti–C, and Ti–O, respectively [40].

In Figure 3g, the functional groups for ZIF-67 and ZIF-67@MXene hybrids were conducted by the Fourier transform infrared spectroscopy (FTIR). The results showed that there was intramolecular association O–H bond (3430 cm^−1^), C–H stretching vibrational peaks (2920 cm^−1^), C=N stretching vibrational peaks (1637 cm^−1^), C–N stretching vibrational peaks (1413, 1135 cm^−1^), tertiary amine nitrogen peaks (1300 cm^−1^), out-of-plane bending vibrations of the imidazole rings (743 cm^−1^), and Co–N vibrational peaks (420 cm^−1^) in the ZIF-67, and the wide OH peak suggested the high association. Moreover, these peaks also emerged in the pattern of ZIF-67@MXene hybrids. In the spectrum of ZIF-67@MXene hybrids, the characteristic peak at 562 cm^−1^ was attributed to the Ti–O stretching vibrational peak, and the peak at 1169 cm^−1^ in the spectrum confirmed the presence of the C–F bond. The appearance of additional bands in the spectrum of ZIF-67@MXene indicated a successful modification.

To study the crystal structure and crystallinity of the as-prepared ZIF-67 and ZIF-67@MXene hybrids, X-ray diffraction (XRD) was measured. Figure 3h shows the crystalline surface of the ZIF-67 as in (002), (112), (022), (013), (222), (114), (233), (224), (134), (044), (334), (244), (235) [37], in addition to the characteristic diffraction peak of ZIF-67, two new diffraction peaks were identified at 2θ = 34.30 and 60.82, corresponding to the (101) and (110) crystallographic planes of MXene. It showed that ZIF-67 and ZIF-67@MXene were prepared successfully. Moreover, it is apparent that the characteristic diffraction peaks of ZIF-67 are maintained after the modification, indicating that the introduction of MXene did not completely interfere with the crystalline phase of ZIF-67. Notably, the sharp and intense characteristic peaks of the ZIF-67 curve reflected the high crystallinity of the as-prepared ZIF-67, while the characteristic peaks of ZIF-67@MXenen hybrids were weaker than those of ZIF-67, indicating that MXene caused the ZIF-67 to present low crystallinity, which verified the indistinctness of the edges of the micromorphology in SEM.

The thermal decomposition processes of ZIF-67 and ZIF-67@MXene hybrids under N_2_ conditions were tested by thermogravimetry and the TG and DTG curves are shown in Figure 3. From the TG curves, the decomposition of ZIF-67 and ZIF-67@MXene hybrids differed greatly. The initial decomposition temperature of ZIF-67 was higher and the thermal stability was better than ZIF-67@MXene hybrids, and then ZIF-67 kept losing weight as the temperature increased, and there was only 40.68 wt% wight left when the temperature was raised to 900 °C. However, ZIF-67@MXene hybrids started to decompose at a much lower temperature. Due to the slow weight loss after 500 °C, the residual char at 900 °C was 68.71 wt%. The results suggested that MXene contributed to the thermal performance of ZIF-67.

### 3.2. Thermogravimetric Analysis of TPU Material

The thermal decomposition process of TPU/MOFs composites under nitrogen and air atmosphere is shown in Figure 4 and detailed data are listed in Table 1. The initial thermal decomposition temperature (*T*_i_) was defined as the temperature at 5% thermal weight loss and *T*_max_ was defined as the temperature at which the thermal weight loss rate peaks.

It can be observed that the TPU composite under nitrogen atmosphere can be divided into two pyrolysis stages. The first stage was the decomposition of the hard TPU segments, where the urethane bonds in the rigid chain segments were degraded to form diisocyanates, alcohols, carbon dioxide, etc. At this stage, the thermal degradation rate of the pure TPU was the greatest among all the composites. The pure TPU decomposed and lost weight at around 300 °C. Its initial decomposition temperature was 317.5 °C and the first peak thermal weight loss rate (*T*_1max_) occurred at 324.8 °C. When ZIF-67 was added to TPU, the *T*_i_ of the composites significantly advanced to 302.0 °C and its *T*_1max_ increased to 327.5 °C, indicating that ZIF-67 can promote the early decomposition of the hard segment of TPU polymer. It is worth noting that *T*_i_ of the TPU/ZIF-67@MXene composite extended to 308.3 °C compared with the TPU/ZIF-67 composites, which indicate that ZIF-67@MXene hybrids can not only promote the early decomposition of the hard segments of the TPU polymer, but also improve the thermal stability of the TPU composites.

The second pyrolysis stage was the decomposition of the soft TPU segment, the decomposition of the soft chain segment polyol into a mixture of polymers, the thermal degradation rate of the pure TPU sample remained maximum at this stage. The temperature increased to 500 °C and the curves became plateaus. The residual char (R_900 °C_) of the pure TPU sample at 900 °C was 3.7%, while residual char of TPU/ZIF-67 and TPU/ZIF-67@MXene composites increased to 7.4 and 7.6%, respectively. The char layer can provide a protective barrier against heat and toxic and harmful gases. Combined with the whole decomposition process, ZIF-67 and ZIF-67@MXene can promote the early decomposition of TPU polymer, showing the catalytic char formation, and ZIF-67@MXenehybrids can enhance the thermal stability of TPU composites significantly.

The degradation of TPU composites under air atmosphere can be divided into three pyrolysis stages. The early initial decomposition temperature of the flame retardant TPU indicated that the addition of ZIF-67 and ZIF-67@MXene hybrids promoted the early decomposition of the TPU. The first stage was the main mass loss stage. Unlike the second stage under nitrogen conditions, where the thermal degradation rates of TPU/ZIF-67 and TPU/ZIF-67@MXene composites were higher than those of the pure TPU, and which decomposed slowly in this stage. The third decomposition stage corresponds to the thermal oxidative degradation of the residual char formed in the previous stage. Therefore, the curve only levelled off when the temperature was increased to 630 °C and all the three samples were almost burnt out with residual char of 0.2%, 1.4%, and 1.6% respectively. This was consistent with the TGA results under nitrogen. Therefore, it can be concluded that IF-67@MXene hybrids contributed to TPU carbon formation during the combustion processes of TPU composites.

### 3.3. Combustion Behavior

The conical calorimeter can effectively test the behavior of polymer composites in terms of heat, smoke, and toxic gas release during the combustion (Figure 4 and Figure 5), revealing the combustion properties of TPU composites, and the relevant data information is listed in Table 2. The data show that the initial ignition time (TTI) of pure TPU material was the longest among all three samples, which demonstrated that the ZIF-67 and ZIF-67@MXene can promote early decomposition of the TPU polymer due to the catalytic effects.

Figure 5a,b shows the heat release rate (HRR) and total heat release (THR) curves with time. It is clear that the peak heat release rate (pHRR) of TPU without flame retardant was the highest. The pHRR value of the TPU material was reduced from 682.25 Kw m^−2^ to 532.61 Kw m^−2^ when ZIF-67 was introduced into TPU, whereas the total heat release (THR) was reduced slightly, therefore, the effect of ZIF-67 on heat reduction is limited. In addition, the pHRR and THR of TPU/ZIF-67@MXene composites were significantly lower by 26% and 9% than those of pure TPU, respectively. Those suggest that ZIF-67@Mxene hybrids are efficient additive flame retardants and can act as barriers against heat transport in TPU. Thus, TPU/ZIF-67@Mxene composites have a low fire risk.

Generally, smoke is one of the most dangerous elements in fire and many fire victims die of smoke asphyxiation. With the addition of the flame retardant, the smoke release rate (SPR) of TPU was reduced markedly as shown in Figure 5c,d. The peak SPR value of TPU/ZIF-67@MXene composites was reduced by 50% compared with the pure TPU. Although the THR of ZIF-67 was hardly reduced, its total smoke release (TSP) value was decreased by 22%, demonstrating the better smoke suppression of ZIF-67. Meanwhile, the TSP value of TPU/ZIF-67@MXene composites was similar to that of TPU/ZIF-67 composites.

As shown in Figure 6a,b, the two main toxic and hazardous gases CO and CO_2_ were investigated. The CO production of TPU composites is another fatal problem, the CO releasing curves show that the ZIF-67 and ZIF-67@MXene can efficiently decrease the CO yield of TPU composites. In recent years, there has been an ongoing push for carbon peaking and carbon neutrality, whose heart of the issue was CO_2_. It was not surprising to find that the CO_2_ yield of TPU/ZIF-67 and TPU/ZIF-67@MXene composites was lower compared with pure TPU, because the inorganic metal part of the ZIF-67 was a good inhibitor of the release of harmful gases.

Moreover, the amount of residual mass is also a relatively important indicator for evaluating flame retardancy. Compared with pure TPU, the residual mass of TPU/ZIF-67@MXene composites is much higher in Figure 7, which is consistent with the results of TGA. ZIF-67@MXene hybrids can catalyze the formation of char layer to exploit the flame-retardant mechanism of the condensed phase, resulting in improved flame retardant properties of TPU/ZIF-67@MXene composites.

### 3.4. Structure and Morphology

The structure and morphology of the char layer play an important role in the flame retardancy of the condensed phase. Digital photographs (top and side views) and SEM photographs of the residual char after CCT are shown in Figure 8. It is obvious that the residual char of the TPU/ZIF-67@MXene composites became swollen, which increased the release path of gas and heat. Thus, its flame retarding effect was evidenced by the fluffy char layer on the surface. Furthermore, the microform of the TPU/ZIF-67@MXene residual char was the flattest among the three samples.

The degree of graphitization of the residual char after CCT was investigated by Raman spectroscopy, the D and G peaks were Raman characteristic peaks of C-atom crystals around 1365 cm^−1^ and 1595 cm^−1^, respectively. The D-peak represents a defect in the C-atom crystal, the G-peak indicates the in-plane stretching vibration of the C-atom sp^2^ hybridization, and the fitted area ratio (ID/IG) characterizes the degree of graphitization. In general, the lower the I_D_/I_G_ value, the higher the degree of graphitization and the higher the strength of the carbon layer. As shown in Figure 9, the TPU/ZIF-67@MXene composites had the lowest I_D_/I_G_ value, dropping from 1.71 in the pure sample to 1.60, indicating the increased degree of graphitization. The highly thermally stable char layer of TPU/ZIF-67@MXene composites can act as a physical barrier, which blocked the transfer of heat and combustible gases. Thus, ZIF-67@MXene mainly played the role in the condensed phase during the combustion process. 

### 3.5. Flame Retarding Mechanism

To further investigate the flame retardant mechanism of TPU composites, the XPS of the carbon slag was measured as shown in Figure 10. In Figure 10a, the pure TPU samples consisted of the elements C, O, and N. In addition to the above elements, TPU/ZIF-67 and TPU/ZIF-67@MXene composites had the element Co on the coke surface, and TPU/ZIF-67@MXene composites also contained Ti element due to the existence of MXene. The corresponding high-resolution spectra of TPU/ZIF-67@MXene composites for Ti 2p, Co 2p, N 1s, C 1s, and O 1s are shown in Figure 10a–c. The Ti 2p spectrum split into four peaks at 465.0, 459.0, 462.8, and 461.1 eV, corresponding to the TiO_2_ bond, Ti^3+^, and Ti^2+^, respectively, which are due to the oxidation of the MXene lamellae by burning to form TiO_2_ [41]. The Co 2p peaks were present at 797.4 and 781.6 eV, respectively, which were mainly attributed to Co_3_O_4_; and there was a companion peak next to each of them, indicating that ZIF-67 was oxidized to porous cobalt oxide nanoparticles. In Figure 9, the N1s spectrum has two peaks at 399.9 and 398.5 eV for Pyrrolic N and Pyridinic N. The characteristic peaks of C 1s were assigned to C=C/C–C, C=N, and C-N bonds at 288.9, 286.2, and 284.8. Furthermore, the two characteristic peaks for O 1 s are C–O (533.5 eV) and C=O/Ti–O bonds (530.8 eV). 

Combining the TG, SEM, Raman, and XPS results, a possible flame retardant mechanism for TPU/ZIF-67@MXene composites is proposed, as shown in Figure 11. As for the gas-phase flame retardant mechanism, the non-flammable gases (H_2_O, CO_2_, N_2_, and NH_3_) were generated during the decomposition process of TPU/ZIF-67@MXene composites and can act as dilution in the combustion region. As for the condensed flame retardant mechanism, ZIF-67 was oxidized and formed porous cobalt oxide during the combustion; and the porous structure complicated the gas and heat transport pathways, inhibiting the exchange of gas and heat. In addition, the lamellar structure of MXene nanosheets can play a physical barrier role, and it can be oxidized to form TiO_2_ during the combustion process. TiO_2_ and Co_3_O_4_ had Lewis acidic sites, they can act as a solid acid to catalyze the degradation and char formation of TPU polymer, resulting in a high-quality protective char layer. The protective char layer can play a physical barrier role in the condensed phase and improved the flame retardant performance of TPU/ZIF-67@MXene composites.

## 4. Conclusions

In this work, a novel ZIF-67-modified MXene hybrid was prepared successfully and incorporated into TPU through melt internal mixing. The SEM results indicated that ZIF-67 was uniformly dispersed on the surface of the MXene sheet layer, and the modification resulted in smaller size of ZIF-67 due to the interaction. The results show that residual char of TPU/ZIF-67@MXene composites was increased obviously compared with pure TPU and the thermal stability of TPU/ZIF-67@MXene composites was better than that of TPU/ZIF-67 composites. When the content of ZIF-67@MXene was only 0.5 wt%, the pHRR, THR, PSPR, and TSP of TPU/ZIF-67@MXene composites were significantly reduced by 26%, 9%, 50%, and 22%, respectively. Meanwhile, the CO and CO_2_ release are also dramatically reduced. Furthermore, the gas phase and condensed phase analyses showed that the improvement in flame retardancy was mainly due to the condensed phase flame retardancy mechanism. The Lewis acid site of TiO_2_ and Co_3_O_4_ can catalyze the formation of a highly thermally stable char layer because the physical barrier effect of MXene can inhibit the exchange of gas and heat between the TPU matrix and the outside world and reduce the heat and smoke release. In summary, this work opened new avenues for the design of high-performance polymeric nanocomposites with excellent fire performance.

## Figures and Tables

**Figure 1 nanomaterials-12-01142-f001:**
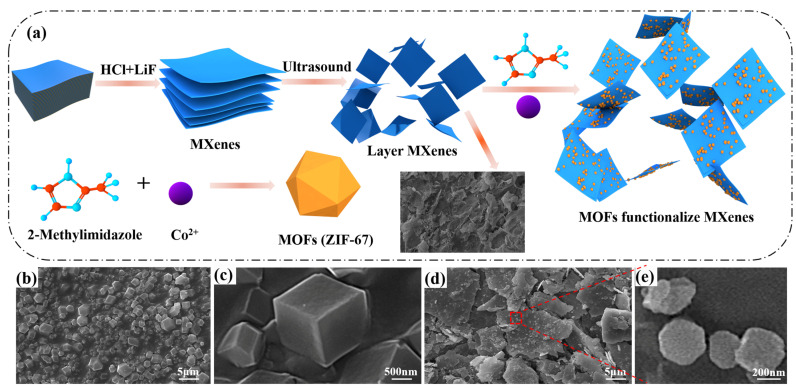
(**a**) Schematic for synthetic route of ZIF-67@MXene hybrids; SEM images of the residues for (**b**,**c**) ZIF-67 and (**d**,**e**) ZIF-67@MXene hybrids.

**Figure 2 nanomaterials-12-01142-f002:**
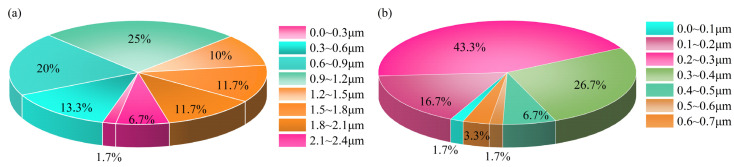
Grain size percentage chart for (**a**) ZIF-67 and (**b**) ZIF-67@MXene hybrids.

**Figure 3 nanomaterials-12-01142-f003:**
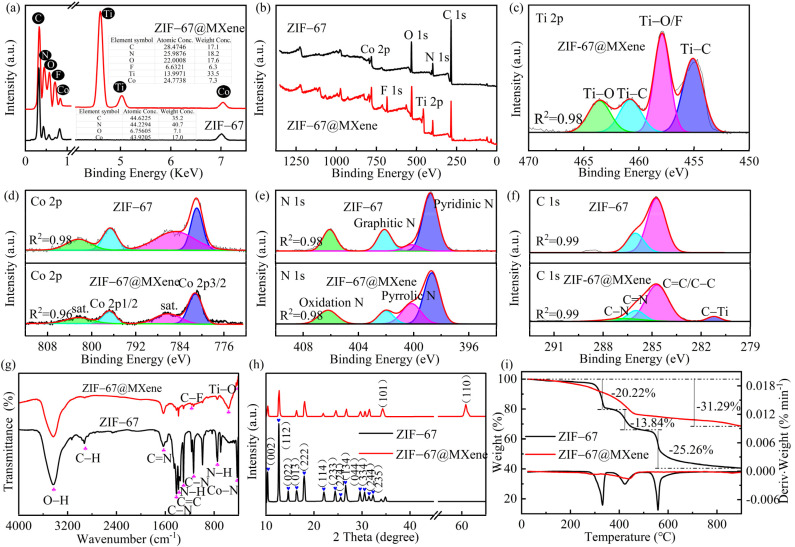
(**a**) EDS; (**b**) XPS survey spectra; the high-resolution (**c**) Ti 2p, (**d**) Co 2p, (**e**) N 1s, and (**f**) C 1s XPS spectra; (**g**) FTIR; (**h**) XRD; (**i**) TG and DTG of ZIF-67 and ZIF-67@MXene.

**Figure 4 nanomaterials-12-01142-f004:**
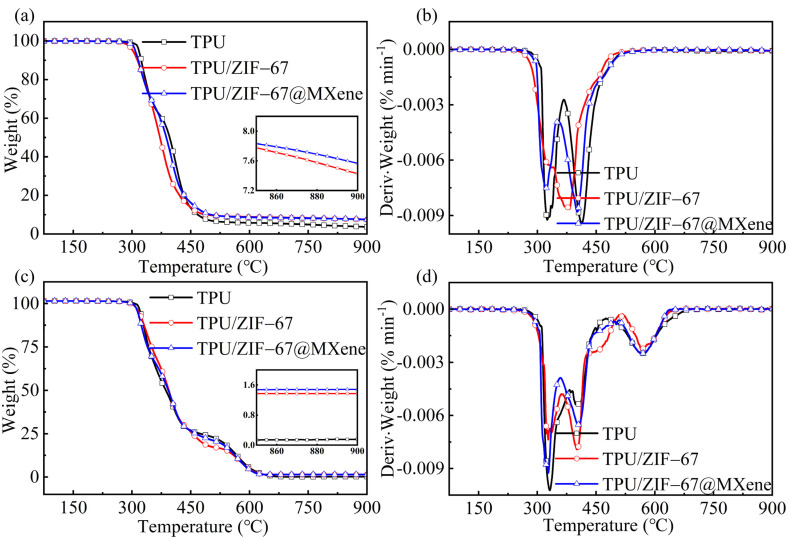
(**a**) TGA and (**b**) DTG curves of TPU composites under nitrogen atmosphere and (**c**) TGA and (**d**) DTG curves of TPU composites under air atmosphere.

**Figure 5 nanomaterials-12-01142-f005:**
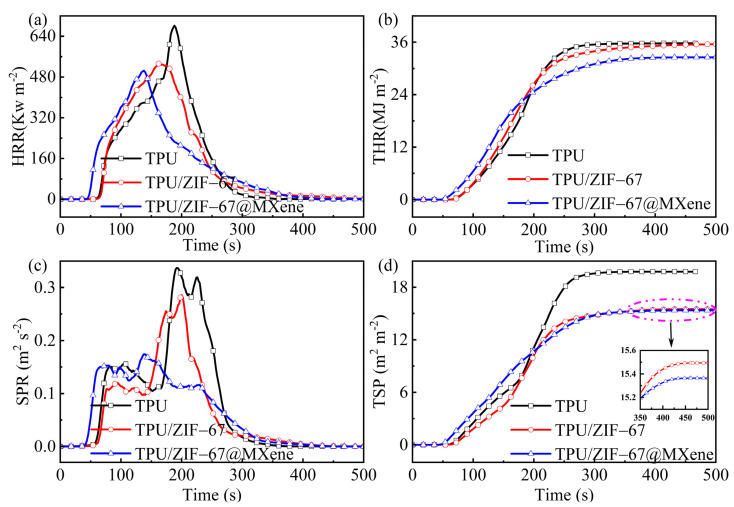
(**a**) HRR, (**b**) THR, (**c**) SPR, and (**d**) TSP yield curves of TPU composites.

**Figure 6 nanomaterials-12-01142-f006:**
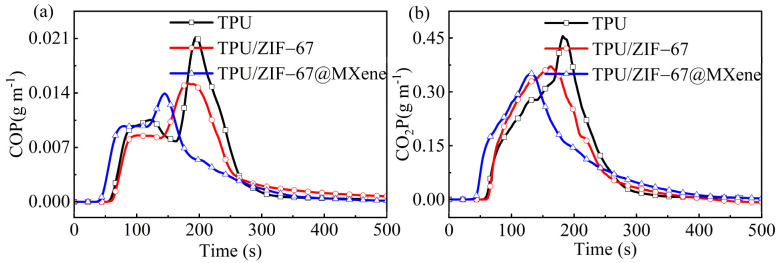
(**a**) CO and (**b**) CO_2_ yield curves of TPU composites.

**Figure 7 nanomaterials-12-01142-f007:**
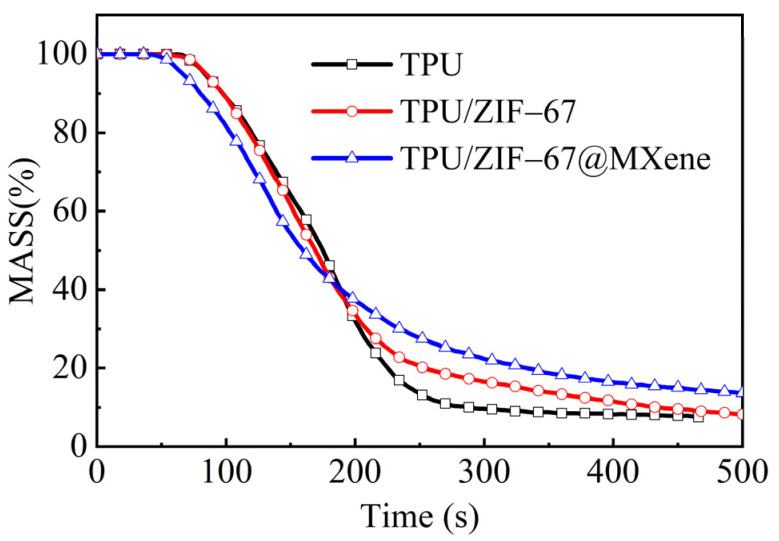
MASS curves of TPU composites.

**Figure 8 nanomaterials-12-01142-f008:**
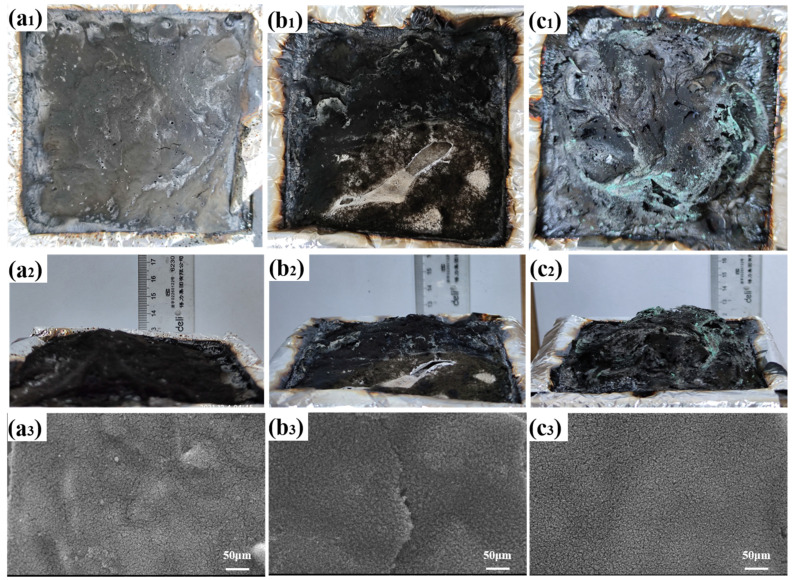
Digital and SEM images of the residues for (**a_1_**–**a_3_**) TPU, (**b_1_**–**b_3_**) TPU/ZIF-67, and (**c_1_**–**c_3_**) TPU/ZIF-67@MXene composites.

**Figure 9 nanomaterials-12-01142-f009:**
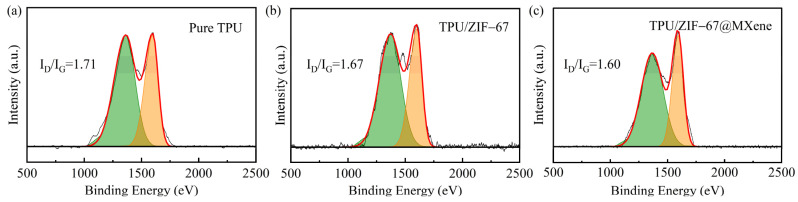
Rama spectra of residual char of pure TPU (**a**), TPU/ZIF-67 (**b**), and TPU/ZIF-67@MXene (**c**) composites.

**Figure 10 nanomaterials-12-01142-f010:**
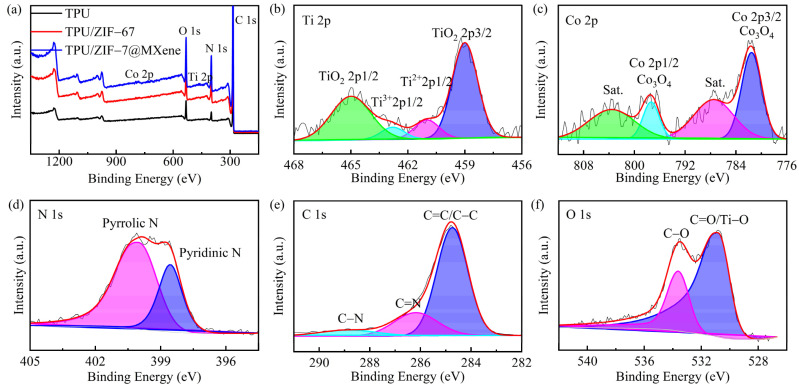
(**a**) XPS survey spectra of the residues of TPU composites; the high-resolution (**b**) Ti 2p, (**c**) Co 2p, (**d**) N 1s, (**e**) C 1s, and (**f**) O1s XPS spectra of the residues of TPU/ZIF-67@MXene composites.

**Figure 11 nanomaterials-12-01142-f011:**
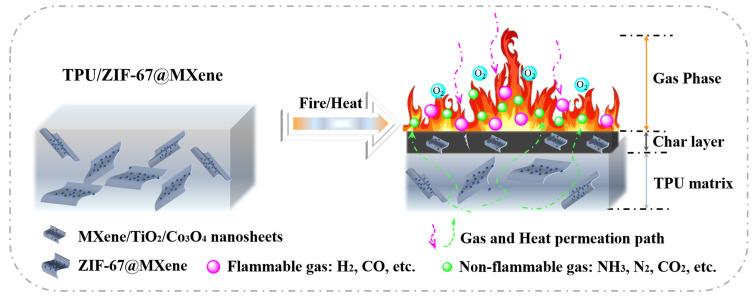
Possible flame-retardant mechanism of TPU/ZIF-67@MXene composites.

**Table 1 nanomaterials-12-01142-t001:** Thermogravimetric analysis data of TPU composites.

Sample	Nitrogen				Air				
	*T*_i_ (°C)	*T*_max_ (°C)		R_900 °C_ (wt%)	*T*_i_ (°C)	*T*_max_ (°C)			R_900 °C_ (wt%)
		*T* _1max_	*T* _2max_			*T* _1max_	*T* _2max_	*T* _3max_	
Pure TPU	317.5	324.8	413.5	3.7	321.3	331.8	398.0	567.3	0.2
TPU/ZIF-67	302.0	327.5	381.5	7.4	317.5	327.8	401.0	565.0	1.4
TPU/ZIF-67@MXene	308.3	324.0	402.5	7.6	314.5	327.8	404.3	567.5	1.5

**Table 2 nanomaterials-12-01142-t002:** CCT data of TPU composites.

Sample	Pure TPU	TPU/ZIF-67	TPU/ZIF-67@MXene
TTI (s)	61	55	45
PHRR (kW m^−2^)	682.25	532.61	505.53
*T*_PHRR_ (s)	188	162	138
THR (MJ m^−2^)	35.75	35.54	32.57
SPR (m^2^ s^−2^)	0.34	0.29	0.17
TSP (m^2^ m^−2^)	19.80	15.50	15.36
Y_CO_ (g s^−1^)	0.21	0.015	0.014
Y _CO2_ (g s^−1^)	0.46	0.37	0.35
MASS (%)	7.53	8.21	13.73

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
