# Peer review of "Metal-organic Framework ZIF-67 Functionalized MXene for Enhancing the Fire Safety of Thermoplastic Polyurethanes"

_nanomaterials, 2022, doi:10.3390/nano12071142_

Round 1
Reviewer 1 Report
Titel:
- The title reads rather bulky and is not fully correct from the English. Please revise and simplify.
Abstract:
- The abstract should be improved. It should give a brief motivation for the conducted work followed by a very brief summary of the methodology.
- Please avoid words like “obviously”.
- The abstract needs to be carefully revised from the language.
- Please rewrite the abstract.
Keywords:
- The selection of the key words is appropriate.
Introduction:
- When using “et al”, please use it in the following way “et al.”
- The background of MXene should be better introduced. A number of recently published, high impact papers are missing. In this regard, you may refer to:
- -- Ten Years of Progress in the Synthesis and Development of MXenes
- -- The rise of MXenes
- -- 2D MXenes: Tunable Mechanical and Tribological Properties
- -- Mechanical Performance of Binary and Ternary Hybrid MXene/Nanocellulose Hydro‐and Aerogels–A Critical Review
- --Two-dimensional transition metal carbides
- -- Two‐Dimensional Nanocrystals Produced by Exfoliation of Ti3AlC2
- The novelty of the study should be better worked out.
Experimental section:
- How does the mechanical mixing to fabricate the TPU composite affect the quality of the hybrids?
- Please provide more experimental details related to the used characterization techniques.
Results and discussion:
- The quality of Figure 1, especially (c) and (e), should be improved.
- The red rectangle remains unclear in Figure 1(d).
- The article misses a full characterization of the generated MXenes. It remains unclear whether the authors work with multi-layer, few-layer, single-layer MXenes? X-y size dimensions should be verified.
- Authors are encouraged to work out the size distribution of the MOF.
- Details in Figure 2a are too small and barely readable.
- XRD results should be presented in a better way.
- In order to fully discuss the characterization results, authors need to characterize the pristine MXene sample. The authors need to include these data in their analysis. Otherwise, you just show half of the reference data.
- Authors are encouraged to better work out and explain why XPS verified the successful fabrication of the hybrid material. This remains unclear and questionable.
- Authors just show a full characterization of the hybrid material, but not of the polymer with the hybrid. Therefore, the characterization necessary appears to be incomplete.
- In Figure 3, only minor changes can be seen. How significant are these changes?
- If possible, authors are encouraged to present all experimentally derived data with mean values and error bars.
- The peak fitting of the Raman measurements is not appropriate. The fitted curves do not represent the experimental data in a sufficient manner. How about the Raman data prior to the combustion experiments? Without these data, it is impossible to draw significant conclusions.
- The quality of the XPS data is rather poor.
- The entire manuscript is rather a description of experimental results, but does not really contain an in-depth discussion. References are not used to back up the claims made by the authors. The discussion part needs to improved significantly to improve the quality of this paper and to make it acceptable for publication.
- References: The presented references are to a certain extent biased. None of the initial, fundamental papers published by Gogotsi or Anasori et al. have been cited. In this sense, the most important articles are not considered/cited in this work related to MXenes.
Author Response
Dear Editors and Reviewers:
Thank you for your letter and for the reviewers’ comments concerning our manuscript entitled “Metal-organic Framework ZIF-67 Functionalized MXene for Enhancing the Fire Safety of Thermoplastic Polyurethanes” (nanomaterials-1599458). Those comments are all valuable and very helpful for revising and improving our paper, as well as the important guiding significance to our research. We have studied comments carefully and have made correction which we hope meet with approval. Revised portion are marked in red in the paper. The main corrections in the paper and the response to the reviewer’s comments are as flowing:
Responds to the reviewer’s comments:
Reviewer 1
- Response to comment: The title reads rather bulky and is not fully correct from the English. Please revise and simplify.
Response:
Considering the Reviewer’s suggestion, we have revised the title.
- Response to comment: Abstract: (1) The abstract should be improved. It should give a brief motivation for the conducted work followed by a very brief summary of the methodology. (2) Please avoid words like “obviously”. (3) The abstract needs to be carefully revised from the language. (4) Please rewrite the abstract.
Response:
Thanks for these reminds. We have revised them.
- Response to comment: Introduction: (1) When using “et al”, please use it in the following way “et al.” (2) The background of MXene should be better introduced. A number of recently published, high impact papers are missing. In this regard, you may refer to: -- Ten Years of Progress in the Synthesis and Development of MXenes; -- The rise of MXenes; -- 2D MXenes: Tunable Mechanical and Tribological Properties; -- Mechanical Performance of Binary and Ternary Hybrid MXene/Nanocellulose Hydro‐and Aerogels–A Critical Review; --Two-dimensional transition metal carbides; -- Two‐Dimensional Nanocrystals Produced by Exfoliation of Ti3AlC2. (3) The novelty of the study should be better worked out.
Response:
Thanks for those comments, I have read literature provided by the reviewers and referred to some relevant literature. As a result, we have modified and supplemented the introductory section of the article.
- Response to comment: Experimental section: (1) How does the mechanical mixing to fabricate the TPU composite affect the quality of the hybrids? Please provide more experimental details related to the used characterization techniques.
Response:
Thanks for this comment. We have apologized that no experimental characterization has been carried out in this paper, while a large of literatures show that lamellar materials such as graphene are prepared by mixing in a similar way, and that mechanical mixing has little effect on the additives.
- Response to comment: Results and discussion: (1) The quality of Figure 1, especially (c) and (e), should be improved. (2) The red rectangle remains unclear in Figure 1(d). (3) The article misses a full characterization of the generated MXenes. It remains unclear whether the authors work with multi-layer, few-layer, single-layer MXenes? X-y size dimensions should be verified. (4) Authors are encouraged to work out the size distribution of the MOF. (5) Details in Figure 2a are too small and barely readable. (6) XRD results should be presented in a better way. (7) In order to fully discuss the characterization results, authors need to characterize the pristine MXene sample. The authors need to include these data in their analysis. Otherwise, you just show half of the reference data. (8) Authors are encouraged to better work out and explain why XPS verified the successful fabrication of the hybrid material. This remains unclear and questionable. (9) Authors just show a full characterization of the hybrid material, but not of the polymer with the hybrid. Therefore, the characterization necessary appears to be incomplete. (10) In Figure 3, only minor changes can be seen. How significant are these changes? (11) If possible, authors are encouraged to present all experimentally derived data with mean values and error bars. (12) The peak fitting of the Raman measurements is not appropriate. The fitted curves do not represent the experimental data in a sufficient manner. How about the Raman data prior to the combustion experiments? Without these data, it is impossible to draw significant conclusions. (13) The quality of the XPS data is rather poor. (14) The entire manuscript is rather a description of experimental results, but does not really contain an in-depth discussion. References are not used to back up the claims made by the authors. The discussion part needs to improved significantly to improve the quality of this paper and to make it acceptable for publication.
Response:
Thanks for these comments. We have repeatedly verified this and made some changes.
Question (1): We are very sorry for our negligence. Figure 1 already has been amended and related information of generated MXenes has been added.
Question (2): Thanks for this comment. We have apologized for any confusion in our presentation, the red box in Figure 1d is enlarged to show Figure 1f.
Question (3): Thank you very much for the question. We are apologized for our negligence, the MXene studied in this paper is a monolayer Ti3C2Tx, the relevant content has been added and amended.
Question (4): Thanks for this comment. We have already worked out the size distribution of the MOF and have supplemented this content according to your proposal.
Question (5): We have apologized for our negligence. Figure 2a already has been amended.
Question (6): Thanks for this comment. We have made the revisions. The analyze of XRD is revised.
Question (7): Thank you very much for your suggestion, the aim of this paper is to focus on the effect of MXene on the flame retardant effect of ZIF-67, some MXene studies were not carried out, so I cannot respond in time, we will make additions and extensions to our future research work.
Question (8): Thanks for this comment. We are very sorry for our misrepresentation, and in combination with electron microscopy and other experiments we can prove that ZIF-67 and ZIF-67@MXene were prepared successfully.
Question (9): Thanks for this comment. The aim of this paper is to focus on the characterization of TPU as a flame retardant, therefore TG and CCT were carried out on TPU/ZIF-67 and TPU/ZIF-67@MXene polymers in this paper.
Question (10): Thanks for this comment. Although the changes are not significant, the addition of MXene is contributing to the thermal stability and carbon residue of TPU, and importantly, it is an important reference and guide for further analysis of the microscopic flame retardant mechanism.
Question (11): Thank you for the suggestion, the XPS experiment has shown the correlation coefficient R2, the treatment of all the experimental data in this paper is reasonable, by checking the literature, the relevant experimental data does not show mean values and error bars.
Question (12): Thanks for this comment. We have verified this; peak fitting of the Raman measurements is acceptable by the Cumulative fitted curves. Then, the Raman test in this paper characterizes the degree of graphitization of the carbon slag, thus, Raman data prior to the combustion experiments do not require to be tested.
Question (13): We apologize for this. It is true that the quality of the experimental data is a bit poor. However, the quality of XPS experimental data is influenced by many factors and we have taken the utmost care in the process. The correlation coefficients R2 are all greater than 98%.
Question (14): Thank you for your question, which has been partially added and modified.
Response to comment: References: The presented references are to a certain extent biased. None of the initial, fundamental papers published by Gogotsi or Anasori et al. have been cited. In this sense, the most important articles are not considered/cited in this work related to MXenes.
Response:
Thanks to your questions and help, we have supplemented the relevant references to MXenes and marked them in red in the text.
We appreciate for Editors/Reviewers’ warm work earnestly, and we hope that the corrections will meet with approval.
Once again, thank you very much for your comments and suggestion.
Best regards,
Mei Wan, Congling Shi, Xiaodong Qian, Yueping Qin, Jingyun Jing, Honglei Che

Reviewer 2 Report
The paper is correctly written, and performed study is complete and accurately described.
However, I encourage authors to add more physical insight and structural /property relationships to strengthen their case. At the moment presented draft looks more like a perfect technical report.
Also, many not standard pr new abbreviations ( that are defined later in the text) should be removed from the abstract. The abstract must be clear and understandable for the general public.
Author Response
Dear Editors and Reviewers:
Thank you for your letter and for the reviewers’ comments concerning our manuscript entitled “Metal-organic Framework ZIF-67 Functionalized MXene for Enhancing the Fire Safety of Thermoplastic Polyurethanes” (nanomaterials-1599458). Those comments are all valuable and very helpful for revising and improving our paper, as well as the important guiding significance to our research. We have studied comments carefully and have made corrections which we hope meet with approval. Revised portions are marked in red in the paper. The main corrections in the paper and the response to the reviewer’s comments are as flowing:
Responds to the reviewer’s comments:
Reviewer 2
- Response to comment:
The paper is correctly written, and performed study is complete and accurately described.
However, I encourage authors to add more physical insight and structural/property relationships to strengthen their case. At the moment presented draft looks more like a perfect technical report.
Response:
Thank you very much for your comment. We have supplemented and revised this article.
- Response to comment:
Also, many not standard pr new abbreviations (that are defined later in the text) should be removed from the abstract. The abstract must be clear and understandable for the general public.
Response:
Thank you very much for your suggestion. We have made revisions to this abstract.
We appreciate for Editors/Reviewers’ warm work earnestly, and we hope that the corrections will meet with approval.
Once again, thank you very much for your comments and suggestion.
Best regards,
Mei Wan, Congling Shi, Xiaodong Qian, Yueping Qin, Jingyun Jing, Honglei Che
Round 2
Reviewer 1 Report
Thank you very much for the revised version. The quality of the revised version has certainly improved. Before being able to recommend the acceptance of this article, authors should address the following points thus aiming at improving the scientific soundness of their results and extending the respective discussion of the obtained results.
- The title reads improved.
- The abstract reads improved as well.
- The provided experimental details related to the characterization are still rather poor and need to be extended. Please provide more information about XPS and Raman spectroscopy.
- Authors replied that they have been using mono-layer MXenes. However, no real experimental proof has been presented for this claim.
- How do you explain the observed differences regarding the size distribution in Figure 2? Authors are advised also to present a size distribution of the MXene nano-sheets. Please provide the respective information and extend the discussion.
- Figure 9, the presented ID/IG values can not be correct. The relative intensity of the D and G peaks are fairly similar and authors present a ratio of 2… This cannot be correct.
- XPS peak fitting is still poor and should be improved.
Author Response
Dear Editors and Reviewers:
Thank you for your letter and for the reviewers’ comments concerning our manuscript entitled “Metal-organic Framework ZIF-67 Functionalized MXene for Enhancing the Fire Safety of Thermoplastic Polyurethanes” (nanomaterials-1599458). Those comments are all valuable and very helpful for revising and improving our paper, as well as the important guiding significance to our research. We have studied comments carefully and have made corrections which we hope meet with approval. Revised portions are marked in red in the paper. The main corrections in the paper and the responds to the reviewer’s comments are as flowing:
Responds to the reviewer’s comments:
Reviewer 1
1. Response to comment: The provided experimental details related to the characterization are still rather poor and need to be extended. Please provide more information about XPS and Raman spectroscopy.
Response:
Considering the Reviewer’s suggestion, we have supplemented more information about XPS and Raman spectroscopy.
2. Response to comment: Abstract: Authors replied that they have been using mono-layer MXenes. However, no real experimental proof has been presented for this claim.
Response:
Thanks for this question. The SEM image shows a single layer of MXene in Figure 1, we are apologized other related experiments cannot be supplemented in time due to the COVID-19.
3. Response to comment: Introduction: (1) How do you explain the observed differences regarding the size distribution in Figure 2? (2) Authors are advised also to present a size distribution of the MXene nano-sheets. Please provide the respective information and extend the discussion.
Response:
Thanks for those comments.
Question (1): In Figure 2, The difference in size distribution may be due to the reaction conditions.
Question (1): We are very sorry that the experiment could not be added in time due to the COVID-19 and it will be our work in the future.
4. Response to comment: Figure 9, the presented ID/IG values can not be correct. The relative intensity of the D and G peaks are fairly similar and authors present a ratio of 2… This cannot be correct.
Response:
We are very sorry for our negligence. We have modified and supplemented the Rama section of the article.
5. Response to comment: XPS peak fitting is still poor and should be improved.
Response:
Thank you very much for your question. We apologize for your troubles, and we have modified the XPS peak fit to combine the analysis of avantage, XPS peak, and Origin. As a result, the most appropriate solution is presented in this paper.
We appreciate for Editors/Reviewers’ warm work earnestly, and we hope that the corrections will meet with approval.
Once again, thank you very much for your comments and suggestion.
Best regards,
Mei Wan, Congling Shi, Xiaodong Qian, Yueping Qin, Jingyun Jing, Honglei Che